# Plant Production with Microalgal Biostimulants

**Domenico Prisa** [1],* and **Damiano Spagnuolo** [2]

1    CREA, Research Centre for Vegetable and Ornamental Crops, Via Dei Fiori 8, 51012 Pescia, Italy
2    Department of Chemical, Biological, Pharmaceutical and Environmental Sciences, University of Messina, Salita Sperone 31, 98166 Messina, Italy; damiano.spagnuolo@unime.it
*    Correspondence: domenico.prisa@crea.gov.it; Tel.: +39-0572-451033

**Abstract:** In order to ensure food security worldwide in the face of current climate changes, a higher quality and quantity of crops are necessary to sustain the growing human population. By developing a sustainable circular economy and biorefinery approaches, we can move from a petroleum-based to a bio-based economy. Plant biostimulants have long been considered an important source of plant growth stimulants in agronomy and agro-industries with both macroalgae (seaweeds) and microalgae (microalgae). There has been extensive exploration of macroalgae biostimulants. A lack of research and high production costs have constrained the commercial implementation of microalgal biostimulants, despite their positive impacts on crop growth, development, and yield. The current knowledge on potential biostimulatory compounds from algae, key sources, and their quantitative information has been summarised in the present review. Our goal is to provide a brief overview of the potential for microalgal biostimulants to improve crop production and quality. A number of key aspects will be discussed, including the biostimulant effects caused by microalgae extracts as well as the feasibility and potential for co-cultures and co-application with other biostimulants and biofertilisers. This article will also discuss the current knowledge, recent developments, and achievements in extraction techniques, types of applications, and timings of applications. Ultimately, this review will highlight the potential of microalgal biostimulants for sustainable agricultural practices, the algal biochemical components that contribute to these traits, and, finally, bottlenecks and involved prospects in commercialising microalgal biostimulants.

**Keywords:** algae biostimulants; sustainable agriculture; microalgae; crop nutrition; biofertiliser; crop protection products

## 1. Introduction

Agro-industry practices and future perspectives are of current relevance, but they face two challenges: improving crop quality and yield due to a growing world population and minimising impacts on the environment and human health. Besides the growing demand for organic food and growing environmental awareness, the forthcoming regulatory framework will restrict the use of chemical input [1–3]. In addition, urbanisation, erosion, and the adverse effects of climate change have further complicated the situation by making farmers worldwide produce more with less due to the genetic potential of stable crops being reached and fertile land areas decreasing [4,5]. It is possible to mitigate these issues with biofertilisers and biostimulants, and they offer renewable options for improving crop quality and yield. Biological fertilisers promote plant growth and development by colonising a plant's rhizosphere with microorganisms such as bacteria, fungi, and microalgae and allowing the plant to absorb nutrients such as nitrogen, phosphorus, potassium, and minerals [6,7]. In contrast, biostimulants are resources that improve crop nutrition, stress tolerance, yield, or quality without damaging or even improving the surrounding environment when they are applied in small amounts [8–10]. Biostimulants increase nutrient efficiency and help plants withstand abiotic stresses, thus enhancing both crop quality and yield. A biostimulant is not a biofertiliser, since it does not directly provide nutrients to plants. As a consequence of

modifying the rhizosphere and plant metabolism, they facilitate nutrient uptake, improving nutrient efficiency, tolerance to abiotic stresses, and crop quality [11]. Humic substances (humic acid, fulvic acid, and humins); algae extracts; protein hydrolysates (signalling peptides and free amino acids); and microorganisms (bacteria, yeast, filamentous fungi, and microalgae) are some of the main plant biostimulants [12]. Macroalgae extracts contain a wide variety of biostimulatory compounds, including amino acids, polysaccharides, vitamins, fatty acids, minerals, phenolics, and phytohormone traces. In the organic plant biostimulant market, macroalgae have been heavily exploited since the early 1980s for their biostimulant potential. In comparison to microalgae, macroalgae have a longer history as biostimulants. The specific modes of biostimulant action of individual bioactivities are often unclear due to their diverse compositions and physicochemical properties. Increasing biotechnological advances, including high-throughput phenotyping and -omic platforms, can be used to illustrate the underlying mechanisms of biostimulant action and to develop novel products [12]. As mainly single-celled photosynthetic organisms, microalgae synthesise a wide variety of metabolites using sunlight and carbon dioxide. Currently, microalgae are being explored for use in biofuels, aquaculture, animal feed, the bioremediation of waste, nutraceuticals, pharmaceuticals, and cosmetics [13]. In the case of agricultural applications, microalgae have not been explored as much as macroalgae. According to traditional research, cyanobacteria (blue–green algae) can fix nitrogen in paddy fields and are beneficial for a variety of other crops [14]. Agricultural microalgal biomass is known to act as a biofertiliser and soil conditioner [15,16]. In contrast, living cyanobacteria are known to act as a potential biocontrol agent against plant pathogens through the activation of plant defence enzymes and the production of hydrolytic enzymes and antimicrobial compounds. In addition, by integrating waste nutrients such as wastewater [17] and anaerobic digestion waste (digestive waste) [18] into microalgal biostimulants, unique opportunities can be provided for shaping circular economy platforms. The major emerging concepts identified in a recent literature survey on scientific trends and market opportunities are microalgal bioplastics and biostimulants [19]. Despite this, determining the mechanisms of biostimulant action of specific bioactivities is still a major challenge due to the variability of algae and crops and the interactions between them based on abiotic factors [12]. It is clear, however, that microalgae have huge potential when viewed alongside the promising capabilities of microalgal biostimulants as well as the increasing costs of chemical fertilisers, pesticide resistance, and climate change. Microalgae can help to make agriculture more sustainable and resilient by providing an immense scope. In this review, we will discuss the current state of the research on algal biostimulants, their sources, and the quantitative data thereof [13]. Phytohormones, proteins, amino acids, humic acids, fulvic acids, polysaccharides, antioxidants, vitamins, and enzymes are among the biostimulants that will be covered. A comparison will also be made between physiological effects and quantitative information on established macroalgal biostimulants to determine the best way forward for the microalgal biostimulant industry [14]. The challenge of the near future lies in the development of cost-effective cultivation facilities and technologies for biomass harvesting, optimisation of microalgae metabolism to maximise productivity, and the production of new molecules of interest. All this will allow the potential of these organisms to be utilised in full, supporting the growth of a sustainable economy. The characteristics that make the use of these photosynthetic organisms attractive and advantageous lie in the facts that they have high and continuous growth rates; they have short life cycles and are capable of living in different growth conditions; and most of the resources required to support the biomass production of microalgae do not compete with agriculture, representing an ideal complementary source to traditional agricultural production. Finally, the conversion efficiency of solar energy into the biomass of algal crops, and thus the productivity per hectare, is much higher than that achievable with conventional crops. We will discuss the potential of microalgal biostimulants in improving crop production and quality, including specific biostimulant effects caused by extracts of microalgae, microalgal–algal consortia,

and microalgal–bacterial consortia; extraction techniques; application types; timing; and current regulatory perspectives.

## 2. Crop Production and Quality Can Be Improved with the Use of Microalgae and Algal–Bacterial Consortia

In terms of effect, plant biostimulants are considered borderline products because they show intermediate effects between fertilisers and plant protection products. Currently, these products are governed by national laws, and they are referred to differently in different European Member States [20], resulting in unfair competition between stakeholders. A common agricultural practice is fertilisation. Microalgae are regarded as organic fertilisers, as they can gradually release N, P, and K to prevent nutrient losses [21,22]. Furthermore, microalgae can recover nitrogen and phosphorus from wastewater by concentrating these nutrients in their biomasses. There are various organic and inorganic compounds in the *Arthrospira* spp., suggesting that these microalgae could be used as biofertilisers. A soil reservoir can be depleted of nutrients when crops are intensively produced. There are also some nutrients that can be affected by soil conditions, such as pH, salinity, and calcium carbonate ($CaCO_3$), as well as the antagonism between macronutrients and micronutrients [23–25]. In addition to macronutrients, micronutrients are also essential for crop growth and development [26]. As micronutrients are coenzymes for a variety of processes in plant metabolism, their deficiencies can cause severe problems in cell activity, resulting in non-optimal growth and decreased crop yield and quality [23,27]. Studies have shown that microalgae as biofertilisers increase nutrient uptake, biomass accumulation, and crop yield [28–30]. As a result of their potential applications in various agricultural cropping systems, as well as to improve agricultural sustainability, different studies have been conducted to assess the effects of MBFs on various crops, including rice, garlic (*Allium sativum* L.), fenugreek (*Trigonella foenum-graecum* L.), and tomato [31–33]. Research has highlighted the potential of bacterial plant biostimulants, including non-pathogenic bacteria such as *Pseudomonas, Bacillus*, *Azotobacter*, *Serratia*, and *Azospirillum*. Upon seed or root inoculation, *Azospirillum brasilense* promotes the growth of many terrestrial plants and improves the yields of a variety of crops [34]. PGPBs are capable of influencing plant growth either directly or indirectly. Direct mechanisms involve the acquisition of resources or the modification of plant hormone levels. Indirect mechanisms, on the other hand, include the promotion of increased plant resistance to pathogen attacks and the activation of a form of defence called induced systemic resistance (ISR). Consequently, it has been proposed that co-immobilised microalgae and PGPBs are effective means of increasing microalgal populations within confined environments as well as improving the biostimulant potential of algal–bacterial consortia. In the spring and summer, such consortia have resulted in 18.9% and 12.9% weight increases, respectively, in romaine lettuce and 16.5% and 22.7% increases, respectively, in leaf lettuce. In contrast to the control romaine lettuce, the treated romaine lettuce increased its carotenoid content by 26.7% during the summer, suggesting that the algal–bacterial consortia reduced the negative effects caused by excess light and heat, resulting in better light-dependent metabolite development. Compared with the control, the treated romaine lettuce had a 2.5-fold higher antioxidant capacity for algal–bacterial treatment in the summer [10]. Additionally, it has been shown that salt stress and temperature stress increase antioxidant activity; elevated concentrations of guaiacol-specific peroxidase have been observed in French beans [35]. Despite this implying that the plants were stressed due to high summer temperatures, the addition of the biostimulant consortia improved the biomass compared with the control plants. Plant yield can also be improved with the consortia of algal strains without bacteria [36]. A co-culture of *Anabaena cylindrica* and the *Nostoc* sp. produced the highest levels of exopolysaccharide (EPS). Compared with monocultures, this consortium application achieved the highest plant yield in lettuce crops, indicating that the EPS was beneficial [37]. *Chrorophyta* improve growth and nutritional quality, while cyanobacteria aid in N fixation, soil quality, and biocontrol. Accordingly, combining the two strains or more is likely to improve agricultural aspects more than just

one strain will, and specific combinations of synergistic traits can be selected depending on the desired outcome [36]. In order to fully exploit the potential of natural co-cultures, it is important to select the right consortium partners strategically [38]. Therefore, further research into different combinations of algae should be conducted to confirm whether they should be cultivated together or whether mutual benefits will still occur if they are cultured separately and applied simultaneously.

## 3. Microalgal Biomass Production

Microalgae can be produced using wastewater, which recovers nutrients and preserves water for further use [39,40], making it one of the most rapidly growing activities in the world [41]. As a result, microalgae reduce greenhouse gas emissions by sequestering $CO_2$ and nitrous oxide ($N_2O$) from industrial by-products. A variety of systems for producing microalgal biomass have been proposed and are being used both in laboratories and in industries [42,43]. Raceway ponds are among the most commonly used methods. In raceway ponds, the water depth is between 10 and 50 cm, which allows for adequate illumination, and a paddle wheel mixes and circulates the gas/medium. As a result of direct exposure to the air, the growing medium will evaporate, regulating the temperature of the culture medium. *Arthrospira* spp., *Dunaliella* spp., *Anabaena* spp., *Phaeodactylum* spp., *Pleurochrysis* spp., *Chlorella* spp., and *Nannochloropsis* spp. are common microalgae and cyanobacteria grown in this system [44]. Due to better capture of light and optimal use of cultivated space, photobioreactors typically have higher volumetric productivity than open ponds. Raceway ponds, on the other hand, require less energy to mix culture media because they are made with less expensive materials [45]. Photobioreactors have better radiant energy utilisation, reduced gas/liquid mass transfer, better temperature control, and lower microalgae productivity than open ponds. As a strategy to increase the cost-effectiveness of microalgal production, the optimisation of culture media has been suggested [46]. Microalgal production can be made more cost-effective by employing low-cost resources such as nutrient-rich wastewater, agricultural by-products, and inexpensive fertilisers [39,47]. In hydroponics, microalgae usually grow spontaneously, but they are considered a critical point because they can cause nutritional competence and pipeline clogs. In contrast, microalgae, through the biochemical process of photosynthesis, can produce oxygen ($O_2$) for crop roots to respire and grow through nutrient solutions. Microalgae can be co-cultivated without any additional inputs, according to some authors [48]. In addition, Barone et al. [49] have suggested the co-production of tomato plants and microalgae (*Scenedesmus quadricauda* or *Chlorella vulgaris*). Microalgal growth and chemical composition can be affected by a number of factors, including nutrients, light intensity, pH, and electroconductivity. Moreover, the different characteristics of the growing medium, particularly the source and concentration of nitrogen (N), can affect the growth and biochemical compositions of microalgal species [50–52]. Variation of the N source influences *Arthrospira* species' biomass production [53,54]. Meanwhile, the growth phase of the marine microalgae in the *Isochrysis* spp. had a greater impact on gross biochemical composition than did the source of N [38]. Microalgae productivity might be enhanced by biochemical stimulants such as phytohormones and polyamines [55,56]. With the green algae in the *Chlorella* spp., biochemical stimulants were tested for their influence on growth and chlorophyll concentration. These microalgae grew significantly faster and produced greater amounts of proteins, saccharides, and chlorophylls using natural and synthetic auxins [57].

## 4. Microalgae as a New Source of Biostimulants

The agricultural applications of microalgae have long focused on their use as biofertilisers and soil conditioners whose effects on crops are mainly attributable to the improvement of physical, chemical, and biological soil fertility [58]. However, in recent years, numerous studies have shown that the variety of physiological responses in plants following the application of these microbial biomasses cannot solely be attributed to the increases in the

nutrients available to plants but also derives from the action of a wide range of bioactive molecules (e.g., phytohormones, amino acids, vitamins, polysaccharides, carbohydrates, polyamines, polyphenols) that are effective on plants at concentrations considerably lower than those of the macroelements (such as nitrogen, phosphorus, and potassium) contained in biofertilisers [59,60]. The ability of microalgae to produce these bioactive molecules, which plants can absorb and metabolise both foliarly and through the root, and the possibility of improving crop productivity using very small quantities of the product compared to biofertilisers has led the scientific community and companies to take an interest in studying the biostimulant properties of these microalgae [61]. Considering that most of the results in this field have been published in recent years and that very few microalgae- and cyanobacteria-based biostimulant products are available on the market today, we can say that the research in this area, although very promising, is still in its infancy [1,59]. Due to the enormous biodiversity of these microorganisms (it is estimated that only half of the approximately 55,000 existing species of microalgae and cyanobacteria have been described to date) [62], only a small number of strains belonging to a few genera have been investigated for their biostimulant properties to date (Figure 1). Most of the products currently on the market are obtained from the cyanobacterium *Arthrospira platensis* and the green microalgae in the *Chlorella* spp. The *Arthrospira* and *Chlorella* spp. are the two genera most extensively cultivated worldwide for various commercial applications (mainly for the nutraceutical market) and most frequently studied for their biostimulant activities on different plant species, appearing in 49% and 56%, respectively, of the scientific publications in the field related to cyanobacteria and microalgae.

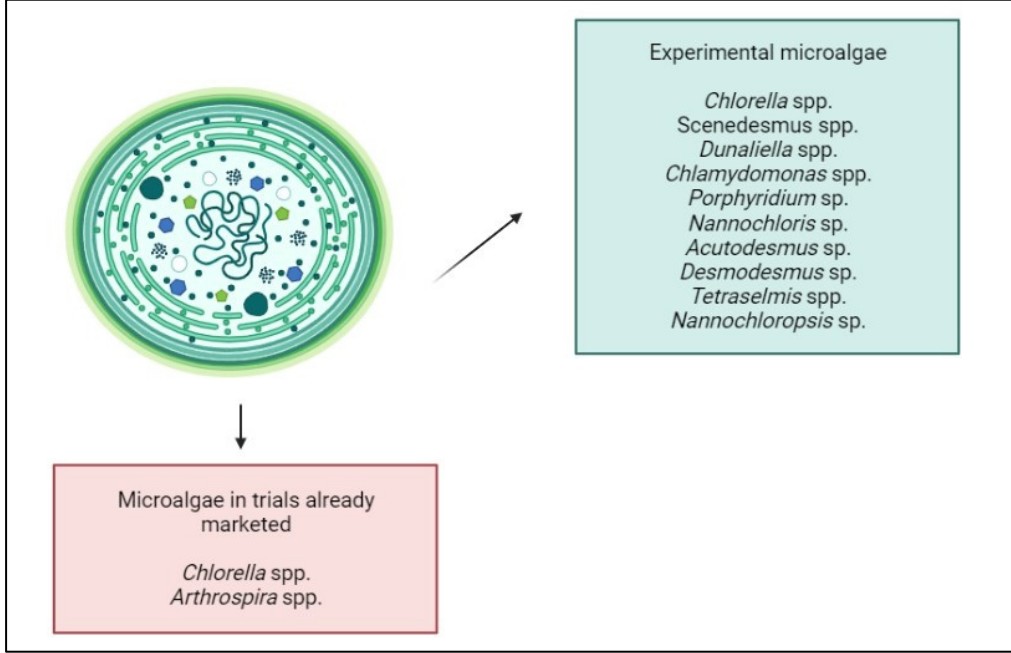

**Figure 1.** Genera of microalgae studied to date for their biostimulant activities with in vivo tests and currently on the market for biostimulant products, listed in descending order by number of published papers.

## 5. Processes and Applications of Biostimulating Algal Biomass

The production of biostimulants from algal biomass and cyanobacteria may involve the use of various techniques mainly aimed at breaking down cells by making bioactive molecules contained in or bound to cell walls available to the plant. This disruption can be achieved through physical/mechanical, chemical, or enzymatic methods [61]. The choice of extraction method is mainly dictated by the type of biomass used and the target molecules. For example, the physical/mechanical methods most commonly used for research purposes

today, which involve mechanical cell wall disruption or the use of high pressure, high temperatures, ultrasound, or combinations thereof, cannot guarantee high extraction yields for micro- and macroalgae, which may have thicker cell walls than cyanobacteria [59]. Cell disruption may be followed by a phase of separation from the extract of cell residues by centrifugation or filtration or with an extraction phase using solvents in order to obtain specific fractions of the crude extract [1]. For instance, in the production of biostimulant polysaccharide extracts, polysaccharides are usually precipitated with ethanol following the physical breakdown of cells. A rather recent technique is extraction that uses supercritical $CO_2$ as a solvent, i.e., with chemical–physical properties intermediately between those of a liquid and those of a gas, obtained at low temperatures (50 °C) and under high pressure (200–500 bars), ensuring the preservation of thermolabile bioactive compounds in biomass [61]. In the preparation of microalgal and cyanobacterial hydrolysates, the use of chemical agents, mainly acids or bases such as sulphuric acid, hydrochloric acid, and sodium hydroxide, generally results in the breakdown of the macromolecules contained in cells. However, these methods have been less and less used, as they may lead to the degradation and inactivation of some bioactive molecules contained in biomass and require the subsequent disposal of large quantities of chemical compounds [63]. Enzymatic methods use single enzymes capable of breaking cell walls and/or proteolytic enzymes that cleave peptide bonds to produce protein hydrolysates, i.e., products rich in free amino acids and soluble peptides. Extracts and hydrolysates can be applied directly to the foliar apparatus by spraying or nebulisation or to the growing medium by fertigation, in which active molecules are absorbed by the root system, or they can be used for pre-sowing treatments [58,61]. Foliar application is generally preferred to soil application, as it allows the use of lower doses of product, limits losses due to leaching, and prevents degradation by soil microorganisms. To avoid the costs associated with the extraction/hydrolysis processes, live cells can be applied directly into growth media or onto plant leaves or be used for seed treatment. Alternatively, culture media separated from microalgal biomass by filtration or centrifugation can be used directly for biostimulant treatment, exploiting the action of the compounds released by the microbial cultures [59].

## 6. Main Biostimulating Effects of Microalgae on Plants

The application of microalgae and cyanobacteria or formulations derived from them (biomass, extracts, hydrolysates) on plants has been shown to produce a wide range of often interconnected beneficial effects (Table 1). These responses vary depending on the microalgal species used to produce each biostimulant but also in relation to the plant species treated and the growing conditions (Figure 2) [64,65]. The phenological stage of the plant and the environmental conditions can also directly influence the success of the biostimulant treatment. Among the most common effects observed is an increase in growth and, consequently, yield in leafy vegetables (lettuce, spinach, rocket) and herbs (mint, basil) [59,64]. These increases in plant growth and fresh weight have been associated with a stimulation of the nitrogen and carbon metabolism in plants treated with microalgal extracts, whereby increases in leaf, protein, carbohydrate, and photosynthetic pigment (chlorophyll and carotenoid) content were observed [66]. The stimulation effects of the primary metabolism can be attributed to an increase in the nutrient uptake of plants subjected to the biostimulant treatment. In this sense, biostimulants can act either directly by improving soil structure and nutrient availability in the soil when applied basally or by directly influencing plant physiology when applied basally or foliarly [67]. Indeed, it is known that the inoculation of cyanobacteria into soil can promote the uptake of zinc and iron by plants through the production of siderophores [68]. In addition, the extracellular polysaccharides produced by many cyanobacterial species can stimulate rhizosphere microbiota by providing organic carbon for microbial growth and can improve soil aggregation and water retention capacity, increasing the volume of soil that can be explored by roots and indirectly promoting root growth [69]. Stimulation of root growth and development has also been observed in several studies after treatment with extracts and hydrolysates of mi-

croalgae and cyanobacteria [1,59]. For example, the use of *Chlorella vulgaris* and *Scenedesmus quadricauda* on beetroot produced positive effects on root architecture, including increases in the root length and also in the number of lateral roots and thus the root surface area for nutrient uptake [49]. These stimulation effects occurred when the biostimulant both was applied to the basal part of the plant and was absorbed directly by the roots, as well as when it was applied to the leaves and induced a concomitant increase in the macro- and micronutrient content in the plant tissues. In addition to foliar and soil application, seedling growth can also be stimulated following seed treatment in the pre-sowing phase. Seed treatment can also have the effect of increasing germination rates [64]. Due to their ability to accelerate germination and seedling development, stimulate early flowering, and increase numbers of flowers, microalgal and cyanobacterial biostimulants may also have interesting applications found in floriculture [64,70]. For example, aqueous extracts and lyophilised biomass of *Desmodesmus subspicatus* increased germination in vitro and accelerated development in the subsequent transplanting and acclimatisation phase in a greenhouse of the orchid *Cattleya warneri* [71]. The effect of biostimulants on plants does not only result in improved vegetative growth. For example, treatment of vines has resulted in a significant increase in grape yield [64]. Furthermore, the application of biostimulants can trigger biochemical processes that lead to the accumulation of important metabolites that improve the quality characteristics and shelf life of a final product [72,73]. Among these, we can mention an increase in the essential oil content of leaves in peppermint treated with the extracts of *Anabaena vaginicola* and *Cylindrospermum michailovskoense* [74] and an increase in total soluble solid content and reduction in weight loss during the storage of onions treated with extracts of *Arthrospira platensis* [75]. Although the incidence of abiotic stresses, such as drought, salinity, and temperature extremes, is expected to increase in the coming years as climate change intensifies, few strategies are available to date to mitigate the negative effects of such stresses [76]. Many abiotic factors have manifested themselves in plants as osmotic stresses, leading to the accumulation of reactive oxygen species (ROSs) that can cause severe oxidative damage to DNA, lipids, carbohydrates, and proteins [77]. It has been shown that the application of microalgae, cyanobacteria, and formulations derived from them will promote the growth and yield of certain plant species, such as rice, wheat, and tomato, under abiotic stress conditions, inducing an enhancement of the antioxidant defences in the plant tissues [59,64]. It is important to remember that the concentration of a biostimulant and its number of applications are determining factors in the success of treatment and that an increase in dose does not always correspond to an increase in positive effects on a plant [78]. In fact, it has been found in some studies that intermediate dilutions of biostimulants may be more effective in promoting growth and flowering, while application of high doses will usually reduce or even neutralise the effect. Effective doses may vary considerably depending on the plant species treated and the method of application. In general, foliar application is effective at lower concentrations than seed or soil application is [59]. In rice plants inoculated with microalgae, accumulations of phenolic acids and flavonoids have been observed in leaf tissue [60]. In addition, according to recent studies, polysaccharide extracts obtained from different microalgal strains, including *Chlamydomonas reinhardtii*, *Chlorella rokiniana*, *Porphyridium* spp., and *Dunaliella salina*, can increase the activity of antioxidant enzymes such as catalase, peroxidase, and superoxide dismutase in tomato plants subjected to salt stress [72]. Another important function of exopolysaccharides released into the soil is to sequester metal ions and sodium ions, reducing their uptake by plants and stimulating their growth in saline or polluted soils [78,79]. For example, coating maize seeds with *Arthrospira platensis* has led to a reduction of more than 90% of the cadmium absorbed by the roots 12 days after sowing [80,81].

**Table 1.** Effects of microalgae on plant species.

| Species | Genera of Microalgae | Effects | Ref. |
|---|---|---|---|
| Lettuce (*Lactuca sativa* L.) | *Chlorella, Scenedesmus quadricauda, Spirulina platensis* | Improved productivity, antioxidant capacity, and carotenoid content and increased dry matter, chlorophyll, and protein in seedlings. | [82] |
| Maize (*Zea mays* L.) | *Spirulina platensis* | Increased production of caryopses and micronutrient absorption. | [83] |
| Aubergine (*Solanum melongena* L.) | *Spirulina platensis* | Increased vegetative growth and fruit production. | [84] |
| Tomato (*Solanum lycopersicum* L.) | *Acutodesmus dimorphus, Chlorella vulgaris, Scenedesmus quadricauda, Nannochloropsis oculata* | Increased seed germination, crop biomass, root development, and dry matter. Increased sugar and carotenoid content in fruit. | [72,85] |
| Pepper (*Capsicum annuum* L.) | *Spirulina platensis, Dunaliella salina* | Plant growth stimulation and salt stress mitigation in seed germination. | [86] |
| Cucumber (*Cucumis sativus* L.) | *Spirulina platensis* | Improved fresh weight. | [82] |
| Fava (*Vicia faba* L.) | *Spirulina platensis* | Improved protein and amino acid levels of roots and sprouts. | [87] |
| Garlic (*Allium sativum* L.) | *Arthrospira fusiformis* | Increased plant height. | [88] |
| Onion (*Allium cepa* L.) | *Spirulina platensis, Scenedesmus subspicatus* | Increased production, photosynthetic pigments, root development, and sugar and protein content. | [89] |

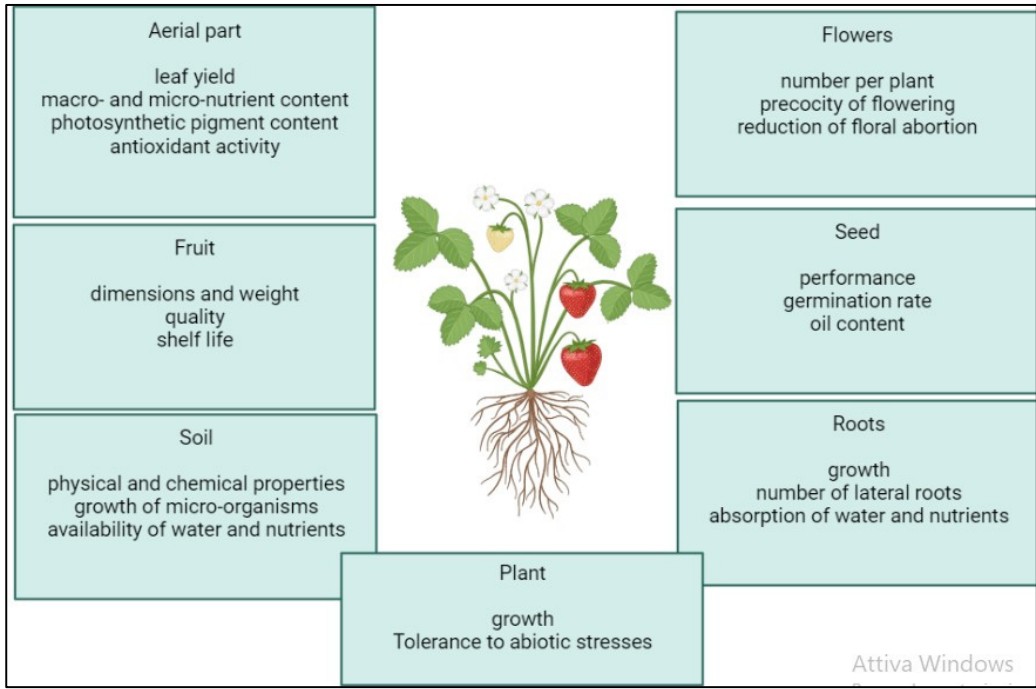

**Figure 2.** Positive effects of the application of microalgal biostimulants on plant growth and physiology, according to the literature: improved aerial growth; improved chemical–physical and biological soil characteristics; increased fruit production and fruit size; increased germination performance and seed oil content; and improved root growth and root hairs with associated increased water and nutrient uptake.

## 7. Use of Microalgal Biostimulants as a Contribution to Sustainable Agricultural Practices

Agriculture consumes a huge amount of nutrients (76 and 87% of the global demand for nitrogen and phosphorus, respectively) to meet the food needs of an actively growing global population [90]. The nutrients that sustain the agricultural system are supplied by unsustainable processes or from sources that are only renewable in the long term. In addition, most fertilisers introduced into agriculture are dispersed into the surrounding

ecosystem (only 17 and 20 percent, respectively, of the nitrogen and phosphorus that enter the agricultural system are then converted into the final products) because, when dispersed into the soil, they complex with organic matter, making it difficult for plants to assimilate them [91,92]. The development of sustainable alternatives for nutrient production to sustain the current agricultural production chain is therefore a priority [93]. Among the alternatives, the high contents of micro- and macronutrients make microalgal biomasses a promising source of biofertilisers [61]. Microalgae can accumulate macronutrients in the form of macromolecules in order to have a reserve and to compensate for the frequent limiting conditions to which they are often subjected in the natural environment. Their high capacity for capturing nutrients from the growth environment also makes them very promising for rehabilitating civil, industrial, and even agricultural wastewater, with the aim of establishing a circular nutrient economy and reducing the environmental impact of an increasingly intensive agricultural practice to meet the needs of a growing population [93]. Various microalgae species have been studied for their applications as biofertilisers with soil-stabilising effects and increased nutrient content as well as increased water retention capacity [94–97]. However, the mechanism responsible for biofertilisation has not yet been fully elucidated. In fact, although microalgae have high nutrient contents, the latter must also be accessible to plants. The biomass of the microalgae could then be degraded by the microbiome of the rhizosphere in order to release constituent nutrients or be subject to natural degradation to allow a sustained release of nutrients. Alternatively, the microalgal biomass could actively interact with plants, inducing the release of bioavailable forms of nitrogen in exchange for carbon compounds from the plant. In the latter case, the microalgae would also have to actively interact with the microbiome of the rhizosphere, and compatibility and survival would therefore not be guaranteed. The compatibility of microalgal biomasses with the rhizosphere microbiome has yet to be systematically investigated. Further studies are needed to establish the biological mechanisms behind the biomass fertilisation activity of microalgae as well as investigating the limitations of implementing this technology at a scale that would reduce the environmental impact of agriculture [70]. In addition to its ability to provide nutrients, the biomasses of microalgae have a wider effect on plant growth through the synthesis of phytostimulant molecules such as hormones [61]. Phytohormones found in extracts from microalgal biomasses include auxins, gibberellin-like molecules, and abscisic acid, with effects on growth and plant development as a result of biostimulatory activity on various metabolic processes, such as photosynthesis, respiration, nucleic acid synthesis, and nutrient assimilation [98]. The application of microalgal biomass extracts also appears to increase resistance to biotic and abiotic environmental stresses; however, the molecular mechanisms underlying this phenomenon remain to be elucidated [93].

## 8. Advantages and Critical Issues in the Use of Microalgae for Biostimulants

Although there is increasing scientific evidence on the beneficial effects of using microalgae (Figure 3), their application in agriculture is still very limited, and very few products are currently on the market, especially when compared to the large number of macroalgae products [1]. One of the main obstacles to be overcome in the commercial exploitation of microalgae relates to the cost of producing biomass, which is generally higher than for macroalgae [59]. Indeed, microalgae are cultivated in controlled and confined systems (photobioreactors and tanks) that require significant amounts of electricity, fertilisers, water, and materials for construction and operation. On the other hand, cultivation in a controlled environment is also one of the main advantages of using microalgal biomasses for biostimulant production, as it allows for greater standardisation of production processes [92]. Macroalgal biomasses, on the other hand, have biochemical and functional characteristics that can vary considerably depending on the phenological stage, environmental conditions, and nutrient availability at the time of harvest, and are therefore more difficult to standardise. In order to make microalgae biostimulants more competitive with other products on the market, it will be necessary to reduce biomass production

costs, e.g., by supplementing cultivation with wastewater treatment, using waste $CO_2$, or cultivating thermotolerant strains that do not require cooling of the crop [88,89]. Ideally, the production of biostimulants from microalgae can be integrated with the production of other products. For example, residual pellets from extraction could be used as a biofertiliser or the remaining lipid fraction could be used for the production of biofuels or to obtain polyunsaturated fatty acids with various cosmetic, medical, and nutraceutical applications or polyhydroxyalkanoates used for the production of bioplastics [59]. Residual proteins could be used to formulate food or feed for animal husbandry and aquaculture. However, the design of an efficient biorefinery system requires that the fractions that contribute most to the biostimulating action be clearly identified in order to assess the possible reuse of the remaining fractions [59].

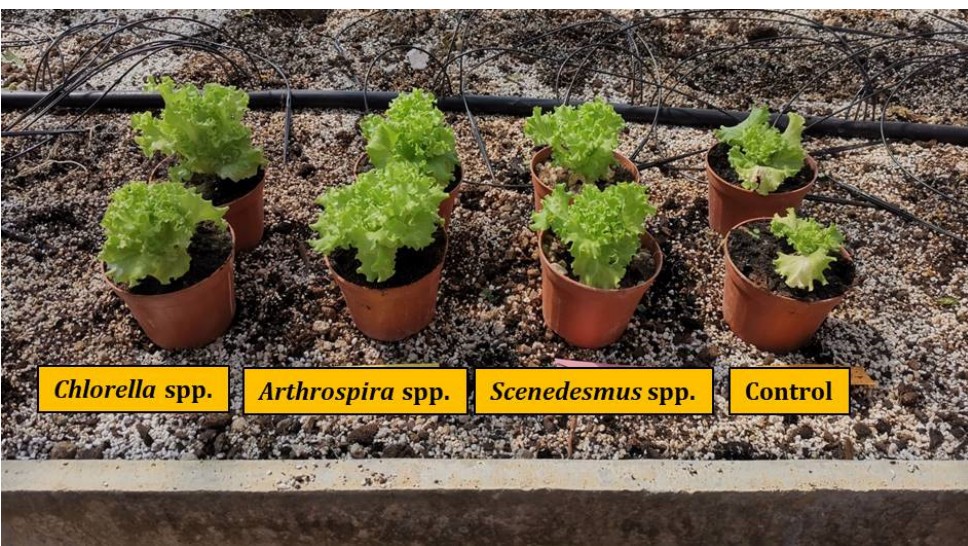

**Figure 3.** Positive effects of the microalgae in the *Chlorella* spp., *Arthrospira* spp., and *Scenedesmus* spp. on the vegetative development of lettuce (*Lactuca sativa* L.).

## 9. Conclusions

Microalgae are, to date, considered one of the most promising sources for the development of biostimulant products, as their wide genetic and metabolic biodiversity is still far from being adequately explored. In recent years, scientific evidence has accumulated to support the biostimulant action of these microorganisms and their derived formulations, which have been shown to increase yield while improving the efficiency of fertilisation treatments by stimulating the nutrient uptake of plants. Furthermore, the application of microalgal biostimulants has been shown to reduce the negative effects on vegetative growth that are caused by abiotic stresses, increasing the resilience of crops to climate change. The success of a biostimulant treatment depends on many factors, including the microalgal species used, the method of biomass processing, the plant species on which it is applied, and the concentration and method of application. It follows that the development of a new biostimulant must involve several stages, ranging from in vitro screening on model species to identify potentially bioactive strains to extensive agronomic testing on different plant species to determine the most convenient and effective doses, timing, and modes of application depending on the species and crop conditions. Although the development and commercialisation of a biostimulant does not currently require a clear demonstration of its mechanism of action, a greater understanding of how different bioactive molecules affect plant physiology would make it possible to accelerate the selection of new biostimulant strains and to plan strategies aimed at increasing the concentrations of bioactive compounds of interest in microalgal biomasses, paving the way for a new category of biostimulants characterised by greater standardisation and reliability and greater efficacy on crops.

**Author Contributions:** Conceptualisation: D.P.; methodology, writing—original draft preparation: D.P. and D.S.; software and investigation, D.P.; writing—review and editing: D.P.; funding acquisition: D.P. All authors have read and agreed to the published version of this manuscript.

**Funding:** This research was funded by the CREA Research Centre for Vegetable and Ornamental Crops.

**Informed Consent Statement:** Not applicable.

**Data Availability Statement:** All data, tables, and figures in this manuscript are original.

**Acknowledgments:** The author would like to express his heartfelt gratitude to his colleagues at the CREA Research Centre for Vegetable and Ornamental Crops in Pescia and to all other sources for their cooperation and guidance in writing this article.

**Conflicts of Interest:** The authors declare no conflict of interest.

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
