# Peer review of "Plant Production with Microalgal Biostimulants"

_horticulturae, doi:10.3390/horticulturae9070829_

Round 1

Reviewer 1 Report

1. Briefly include the mechanisms by which PGPB stimulate plant growth (lines 102.103)

2. Line 307: consider that the phenological stage, and environmental conditions, may also influence in the success of the biostimulant treatment

3. Write in italics the scientific names and terms derived from Latin noted in the text and in the References. For example.

Línea 70: Solanum melongena

Lines 276 y 381: in vitro

Line 427: Lactuca sativa

Lines 485 y 486: Spirulina platensis

Line 489: Chlorella vulgaris

Line 489: Dunaliella primolecta

Line 493: Chlorella pyrenoidosa

Line 519: Arthrospira platensis and Scenedesmus sp.

Line 521: Desmodesmus subspicatus

Line 522. Cattleya warneri.

Lines 524-525: Solanum lycopersicum L.

Line 529: Menthas piperita

Line 530: Allium cepa  and Spirulina platensis

Lines 540, 542, 549, 556: Spirulina platensis

Line 559: Allium sativum

Author Response

Responses to the first reviewer

Good evening I am sending you the changes made according to your suggestions.

  1. Briefly include the mechanisms by which PGPB stimulate plant growth (lines 102.103)

I have included the mechanisms by which PGPBs stimulate plant growth

PGPBs are capable of influencing plant growth either directly or indirectly. Direct mechanisms involve the acquisition of resources or the modification of plant hormone levels. Indirect mechanisms, on the other hand, include the promotion of increased plant resistance to pathogen attacks and the activation of a form of defence called induced systemic resistance (ISR).

  1. Line 307: consider that the phenological stage, and environmental conditions, may also influence in the success of the biostimulant treatment

I have included the part on phenological and environmental phenomena

The phenological stage of the plant and the environmental conditions can also directly influence the success of the biostimulant treatment.

  1. Write in italics the scientific names and terms derived from Latin noted in the text and in the References. For example.

Línea 70: Solanum melongena

Lines 276 y 381: in vitro

Line 427: Lactuca sativa

Lines 485 y 486: Spirulina platensis

Line 489: Chlorella vulgaris

Line 489: Dunaliella primolecta

Line 493: Chlorella pyrenoidosa

Line 519: Arthrospira platensis and Scenedesmus sp.

Line 521: Desmodesmus subspicatus

Line 522. Cattleya warneri.

Lines 524-525: Solanum lycopersicum L.

Line 529: Menthas piperita

Line 530: Allium cepa  and Spirulina platensis

Lines 540, 542, 549, 556: Spirulina platensis

Line 559: Allium sativum

I have corrected all names in italics as indicated

Reviewer 2 Report

The MS  "Plants Production with Microalgal Biostimulants" is an attempr to review studies on the use of microalgae in agriculture as biostimulants and biofertilizers. The subject is very important and urgent, but the MS has some shortcomings. 

Major concerns:

1. The MS devoted to microalgae should probably not begin with potential of bacterial plant biostimulants. The algal-bacteria consortium potential could be added later in the MS.

2. The subtitle of the third section - possibly better Production of microalgae biomass ot Microalgae biomass production. 

3.  Fig 1 shoes that Anthrospira is used commercially, but not studied at all. Is that right? What do arrows mean?

4. Section 5 is poorly structured. It would be useful to divide the effects of (a) extracts and hydrolysates, (b) live cells and (c) culture medium with compounds released by microalgae.  Throught out the whole paper the effects of microalgae as biostimulants and fertilizers are mixed.   

5. Fig. 2. How presumably microalgae may increase water availability?

6. What is the novelty of your review compared to others (ref. 1, 22, 45, 49, 50) and in particular compared to Ronga et al. 2019?

Minor concerns:

Lines 44-47: Please, re-write the sentence as the end is nor clear. 

Line 156: the same or some?

Lines 276, 381 in vitro should be italicazed.

Lines 278-280 Repeated sentence

Line 281: Treatment of wheat could not increase grape yield. 

Table 1: Maize: increased production of micronutrients. Probably you mean that microalgae supply plants with micronutrients.

Latin plant names throughout the text and in the references should be italicazed.

Some sentences are not clear, for example Lines 44-47

Line 156: the same or some?

Please, check the use of prepositions throughout the text. 

Author Response

Responses to the second reviewer

Good evening I am sending you the changes made according to your suggestions.

The MS  "Plants Production with Microalgal Biostimulants" is an attempr to review studies on the use of microalgae in agriculture as biostimulants and biofertilizers. The subject is very important and urgent, but the MS has some shortcomings.

Major concerns:

  1. The MS devoted to microalgae should probably not begin with potential of bacterial plant biostimulants. The algal-bacteria consortium potential could be added later in the MS.

I consider this beginning a distinction from other reviews, where usually little emphasis is placed on these types of interactions

  1. The subtitle of the third section - possibly better Production of microalgae biomass ot Microalgae biomass production.

I have replaced the subtitle as indicated

Microalgae biomass production

  1. Fig 1 shoes that Anthrospira is used commercially, but not studied at all. Is that right? What do arrows mean?

Figure 1 shows that Arthrospira is already used commercially as Chlorella

  1. Section 5 is poorly structured. It would be useful to divide the effects of (a) extracts and hydrolysates, (b) live cells and (c) culture medium with compounds released by microalgae. Throught out the whole paper the effects of microalgae as biostimulants and fertilizers are mixed.

 I find the paragraph more explanatory in this way, so that the reader can get a general idea of the effects of microalgae

  1. Fig. 2. How presumably microalgae may increase water availability?

Figure 2 shows how the use of microalgae can have an effect on root development and their effeciency in absorbing water and nutrients

  1. What is the novelty of your review compared to others (ref. 1, 22, 45, 49, 50) and in particular compared to Ronga et al. 2019?

Compared to the given references, the review places more emphasis on the interactions between algae and bacteria, the effects on plants, especially vegetables and cereals, and the production methods of microalgal biomass.

Minor concerns:

Lines 44-47: Please, re-write the sentence as the end is nor clear.

The sentence has been corrected and rewritten

Biological fertilizers promote plant growth and development by colonizing the plant's rhizosphere with microorganisms such as bacteria, fungi, and microalgae, and allowing the plant to absorb nutrients such as nitrogen, phosphorus, potassium, and minerals

Line 156: the same or some?

some

Lines 276, 381 in vitro should be italicazed.

all words in vitro have been put in italics

Lines 278-280 Repeated sentence

deleted repeated sentence

Line 281: Treatment of wheat could not increase grape yield.

changed the sentence

For example, vine treatments led to a significant increase in grape yields

Table 1: Maize: increased production of micronutrients. Probably you mean that microalgae supply plants with micronutrients.

changed the statement in table 1, where it was intended to indicate an increased absorption of micronutrients

Latin plant names throughout the text and in the references should be italicazed.

All Latin names have been corrected and put in italics

Comments on the Quality of English Language

Some sentences are not clear, for example Lines 44-47

The sentence has been corrected and made more understandable

Line 156: the same or some?

some

Please, check the use of prepositions throughout the text.

All prepositions in the text have been checked

Reviewer 3 Report

The paper titled:"Plants Production with Microalgal Biostimulants", by Prisa and Spagnuolo, is a well-written, and well-structured research, the authors provide a comprehensive review of microalgal biostimulants and their potential in improving crop production and quality. The paper aims to summarize the current knowledge on biostimulatory compounds derived from microalgae, their sources, and quantitative information. Additionally, it discusses the effects of microalgae extracts as biostimulants, the feasibility of co-cultures and co-application with other biostimulants and biofertilizers, extraction techniques, application types, timing, and current regulatory perspectives. The ultimate goal is to highlight the potential for microalgal biostimulants to contribute to sustainable agricultural practices and identify bottlenecks and prospects in commercializing these products. Yet, there is some modifications that must be addressed before considering your work for publication in Horticulturae.

* The introduction could be organized more clearly to guide the reader through the main points. Consider dividing the introduction into paragraphs or sections that correspond to different aspects of the topic, such as the challenges faced by the agro-industry, the role of biofertilizers and biostimulants, the potential of microalgae, and the need for further research.

* While the introduction mentions the limited exploration of microalgae in agricultural applications, it would be helpful to elaborate on the specific gap in the existing literature. What aspects of microalgal biostimulants have been understudied or underutilized? Are there specific challenges or limitations that need to be addressed? 

* Figure 1, can be substituted with a table containing the most important genera, and examples of the most utilized species within each genus or family.

* The authors are invited to add a new section or merge the following suggestion with an existing section in the paper, to emphasize the importance of addressing the challenges faced by the agro-industry in improving crop quality and yield while minimizing environmental impacts.

The authors must clearly articulate how the utilization of microalgal biostimulants can contribute to sustainable agricultural practices and provide a promising alternative to chemical inputs, some recent and relevant references are needed.

Author Response

Responses to the third reviewer

Good evening I am sending you the changes made according to your suggestions.

The paper titled:"Plants Production with Microalgal Biostimulants", by Prisa and Spagnuolo, is a well-written, and well-structured research, the authors provide a comprehensive review of microalgal biostimulants and their potential in improving crop production and quality. The paper aims to summarize the current knowledge on biostimulatory compounds derived from microalgae, their sources, and quantitative information. Additionally, it discusses the effects of microalgae extracts as biostimulants, the feasibility of co-cultures and co-application with other biostimulants and biofertilizers, extraction techniques, application types, timing, and current regulatory perspectives. The ultimate goal is to highlight the potential for microalgal biostimulants to contribute to sustainable agricultural practices and identify bottlenecks and prospects in commercializing these products. Yet, there is some modifications that must be addressed before considering your work for publication in Horticulturae.

* The introduction could be organized more clearly to guide the reader through the main points. Consider dividing the introduction into paragraphs or sections that correspond to different aspects of the topic, such as the challenges faced by the agro-industry, the role of biofertilizers and biostimulants, the potential of microalgae, and the need for further research.

I think the introduction is clear, I have added a part that better defines the possible potential of microalgae in agriculture

The challenge of the near future lies in the development of cost-effective cultivation facilities and technologies for biomass harvesting, optimisation of microalgae metabolism to maximise productivity and the production of new molecules of interest. All this will allow the potential of these organisms to be utilised to the full, supporting the growth of a sustainable economy. The characteristics that make the use of these photosynthetic organisms attractive and advantageous lie in the fact that: they have a high and continuous growth rate; they have a short life cycle and are capable of living in different growth conditions; most of the resources required to support the biomass production of microalgae do not compete with agriculture, representing an ideal complementary source to traditional agricultural production. Finally, the conversion efficiency of solar energy into biomass of algal crops, and thus the productivity per hectare, is much higher than that achievable with conventional crops. It discusses the potential of microalgal biostimulants in improving crop production and quality, including specific biostimulant effects caused by extracts of microalgae, microalgal-algae consortia, microalgal-bacterial consortia, extraction techniques, application types, timing, and current regulatory perspectives.

* Figure 1, can be substituted with a table containing the most important genera, and examples of the most utilized species within each genus or family.

Figure 1 defines in one box the most studied algal genera and in another box those already commercialised

* While the introduction mentions the limited exploration of microalgae in agricultural applications, it would be helpful to elaborate on the specific gap in the existing literature. What aspects of microalgal biostimulants have been understudied or underutilized? Are there specific challenges or limitations that need to be addressed? 

* The authors are invited to add a new section or merge the following suggestion with an existing section in the paper, to emphasize the importance of addressing the challenges faced by the agro-industry in improving crop quality and yield while minimizing environmental impacts.

The authors must clearly articulate how the utilization of microalgal biostimulants can contribute to sustainable agricultural practices and provide a promising alternative to chemical inputs, some recent and relevant references are needed.

I have inserted a new paragraph that better clarifies this with the relevant references

  1. Use of microalgal biostimulants as a contribution to sustainable agricultural practices

Agriculture consumes a huge amount of nutrients (76 and 87% of the global demand for nitrogen and phosphorus respectively) to meet the food needs of an actively growing global population [76]. The nutrients to sustain the agricultural system are supplied by unsustainable processes or from renewable sources only in the long term. In addition, most fertilisers introduced into agriculture are dispersed into the surrounding ecosystem (only 17 and 20 per cent respectively of the nitrogen and phosphorous that enter the agricultural system are then converted into final products) because, when dispersed into the soil, they complex with organic matter making it difficult for plants to assimilate them [77,78]. The development of sustainable alternatives for nutrient production to sustain the current agricultural production chain is therefore a priority [79]. Among the alternatives, the high content of micro- and macro-nutrients makes microalgae biomass a promising source of biofertilisers [47]. Microalgae can accumulate macro-nutrients in the form of macromolecules in order to have a reserve and to compensate for the frequent limiting conditions to which they are often subjected in the natural environment. Their high ca-pacity for capturing nutrients from the growth environment also makes them very promising for rehabilitating civil, industrial and even agricultural wastewater, with the aim of establishing a circular nutrient economy and reducing the environmental impact of an increasingly intensive agricultural practice to meet the needs of a growing population [79]. Various microalgae species have been studied for their application as biofertilisers, with soil stabilising effects and increased nutrient content, as well as increased water re-tention capacity [80-83]. However, the mechanism responsible for biofertilisation has not yet been fully elucidated. In fact, although microalgae have a high nutrient content, the latter must also be accessible to plants. The biomass of the microalgae could then be de-graded by the microbiome of the rhizosphere in order to release constituent nutrients, or be subject to natural degradation to allow a sustained release of nutrients. Alternatively, the microalgae biomass could actively interact with plants, inducing the release of bioa-vailable forms of nitrogen in exchange for carbon compounds from the plant. In the latter case, the microalgae would also have to actively interact with the microbiome of the rhi-zosphere and compatibility and survival is therefore not guaranteed. The compatibility of microalgae biomass with the rhizosphere microbiome has yet to be systematically inves-tigated. Further studies are needed to establish the biological mechanisms behind the biomass fertilisation activity of microalgae as well as to investigate the limitations in im-plementing this technology at scale to reduce the environmental impact of agriculture [56]. In addition to its ability to provide nutrients, the biomass of microalgae has a wider effect on plant growth through the synthesis of phytostimulant molecules such as hormones [47]. Phytohormones found in extracts from microalgae biomass include auxins, gibberellin-like molecules and abscisic acid, with effects on growth, plant development as a result of biostimulatory activity on various metabolic processes such as photosynthesis, respiration, nucleic acid synthesis and nutrient assimilation [84]. The application of mi-croalgae biomass extracts also appears to increase resistance to biotic and abiotic envi-ronmental stresses, however, the molecular mechanisms underlying this phenomenon remain to be elucidated [85].

Round 2

Reviewer 2 Report

The author took into account minor concerns, but 4 out of 6 major concernes were ignored. 

1. It is good that you provide information on interactions, but  why should you begin with this?

2. Accepted

3. Anthrospira is not listed in the box of studied mixroalgae being commercially used and it rised question.

4. I don't mind providing general idea, but don't mix in one direct and indirect effecs of microalgae.

5. It relates to higer water content in plants due to better root work (better water relations), but not water availability (in substrate).

Author Response

Responses to the second reviewer

Good evening I am sending you the changes made according to your suggestions

The author took into account minor concerns, but 4 out of 6 major concernes were ignored.

  1. It is good that you provide information on interactions, but why should you begin with this?

I have changed the title of Chapter 2: Crop production and quality can be improved with the use of microalgae and al-gal-bacterial consortia

And inserted a new part on the effects of biostimulants on plants, in particular with the use of microalgae, with related bibliography. Subsequently the part on the interactions

In terms of effect, plant biostimulants are considered borderline products because they show intermediate effects between fertilisers and plant protection products. Currently, these products are governed by national laws, and they are referred to differently in different European Member States [20], resulting in unfair competition between stakeholders. A common agricultural practice is fertilization. In this regard, microalgae are regarded as an organic fertiliser, as they can gradually release N, P, and K to prevent nutrient losses [21,22]. Furthermore, microalgae can recover nitrogen and phosphorus from wastewater by concentrating these nutrients in their biomass. There are various organic and inorganic compounds in Arthrospira spp., suggesting that this microalga could be used as a biofertilizer. The soil reservoir can be depleted of nutrients when crops are intensively produced. There are also some nutrients that can be affected by soil conditions, such as pH, salinity, calcium carbonate (CaCO3), and antagonism between macronutrients and micronutrients [23,24,25]. Besides macronutrients, micronutrients are also essential for crop growth and development [26]. As coenzymes for a variety of processes in plant metabolism, micronutrient deficiencies can cause severe problems in cell activities, resulting in non-optimal growth and decreased crop yields and quality [23,27]. Studies have shown that microalgae as biofertilisers increase nutrient uptake, biomass accumulation, and crop yields [28,29,30]. As a result of their potential applications in various agricultural cropping systems, as well as to improve agricultural sustainability, different studies were conducted to assess the effects of MBF on various crops, including rice, garlic (Allium sativum L.), fenugreek (Trigonella foenum-graecum L.), and tomato [31,32,33].

  1. La Torre, A.; Battaglia, V.; Caradonia, F. An overview of the current plant biostimulant legislations in different European Member States. J. Sci. Food Agric. 2016, 96, 727–734.
  2. Coppens, J.; Grunert, O.; Van Den Hende, S.; Vanhoutte, I.; Boon, N.; Haesaert, G.; De Gelder, L. The use of microalgae as a high-value organic slow-release fertiliser results in tomatoes with increased carotenoid and sugar levels. J. Appl. Phycol. 2016, 28, 2367–2377.
  3. Mulbry, W.; Kondrad, S.; Pisarro, C. Biofertilisers from algal treatment of dairy and swine manure effluents. J. Veg. Sci. 2007, 12, 107–125.
  4. Shaaban, M.M.; El-Fouly, M.M.; Abdel-Maguid, A.A. Zinc-Boron relationship in wheat plants grown under low or high levels of calcium carbonate in the soil. Pak. J. Biol. Sci. 2004, 7, 633–639.
  5. Shaaban, M.M.; Loehnertz, O.; El-Fouly, M.M. Grapevine genotypic tolerance to lime and possibility of chlorosis recovery through micronutrients foliar application. Int. J. Bot. 2007, 3, 179–187.
  6. Shaaban, M.M.; Hussein, M.M.; El Saady, A.M. Nutritional status in shoots of barley genotypes as affected by salinity of irrigation water. Am. J. Plant Physiol. 2008, 3, 89–95.
  7. Marschner, H. Mineral Nutrition of Higher Plants, 2nd ed.; Academic Pres: Boston, MA, USA, 1995.
  8. El-Fouly, M.M.; Shaaban, M.M.; El-Khdraa, T.F. Soil and plant nutritional status in fruit orchards in Syria. Acta Agron. Hung. 2008, 56, 363–370.
  9. Shaaban, M.M. Nutritional status and growth of maise plants as affected by green microalgae as soil additives. J. Biol. Sci. 2001, 6, 475–479.
  10. Shaaban, M.M. Green microalgae water extract as foliar feeding to wheat plants. Pak. J. Biol. Sci. 2001, 4, 628–632.
  11. Faheed, F.A.; Fattah, Z.A. Effect of Chlorella vulgaris as Biofertiliser on Growth Parameters and Metabolic Aspects of Lettuce Plant. J. Agric. Soc. Sci. 2008, 4, 165–169.
  12. Tarraf, S.A.; Talaat, I.M.; El-Sayed, A.E.K.B.; Balbaa, L.K. Influence of foliar application of algae extract and amino acids mixture on fenugreek plants in sandy and clay soils. Amino Acids 2015, 16, 19–58.
  13. Paudel, Y.P.; Pradhan, S.; Pant, B.; Prasad, B.N. Role of blue green algae in rice productivity. Agric. Biol. J. N. Am. 2012, 3, 332–335.
  14. Shalaby, T.A.; El-Ramady, H. Effect of foliar application of bio-stimulants on growth, yield, components, and storability of garlic (Allium sativum L.). Aust. J. Crop Sci. 2014, 8, 271.

  1. Accepted

  1. Anthrospira is not listed in the box of studied mixroalgae being commercially used and it rised question.

I have changed figure 1 to experimental microalgae and microalgae in trial that are already commercialised

  1. I don't mind providing general idea, but don't mix in one direct and indirect effecs of microalgae.

I have changed the order of the sentences within the paragraph to better divide the direct actions of microalgae from those that occur indirectly on plants

It is important to remember that the concentration of the biostimulant and the number of applications are a determining factor in the success of the treatment and that an increase in the dose does not always correspond to an increase in the positive effects on the plant [81]. In fact, it has been found in some studies that intermediate dilutions of biostimulant may be more effective in promoting growth and flowering, while the application of high doses usually reduces or even neutralises the effect. Effective doses may vary considerably depending on the plant species treated and the method of application. In general, foliar application is effective at lower concentrations than seed or soil application [59].In rice plants inoculated with microalgae, an accumulation of phenolic acids and flavonoids has been observed in leaf tissue [60]. In addition, according to recent studies, polysaccharide extracts obtained from different microalgal strains, including Chlamydomonas reinhardtii, Chlorella rokiniana, Porphirydium spp. and Dunaliella salina, can increase the activity of antioxidant enzymes such as catalase, peroxidase and superoxide dismutase in tomato plants subjected to salt stress [72]. Another important function of exopolysaccharides released into the soil is to sequester metal ions and sodium ions, reducing their uptake by plants and stimulating their growth in saline or polluted soils [78,79]. For example, coating maize seeds with Arthrospira platensis led to a reduction of more than 90% of the cadmium absorbed by the roots 12 days after sowing [80].

  1. It relates to higer water content in plants due to better root work (better water relations), but not water availability (in substrate).

I have added in the caption of figure 2 all the explanations related to the photo, with reference also to the root system where the algae stimulate the growth of the roots with a related increase in water and nutrient uptake.

Reviewer 3 Report

The authors have taken into account all of my comments in their manuscript, making it appropriate for publication in its current state.

Author Response

Responses to the third reviewer

Good evening, as suggested by the second reviewer I am sending you the changes made to the document according to the new suggestions. Thank you

  1. It is good that you provide information on interactions, but why should you begin with this?

I have changed the title of Chapter 2: Crop production and quality can be improved with the use of microalgae and al-gal-bacterial consortia

And inserted a new part on the effects of biostimulants on plants, in particular with the use of microalgae, with related bibliography. Subsequently the part on the interactions

In terms of effect, plant biostimulants are considered borderline products because they show intermediate effects between fertilisers and plant protection products. Currently, these products are governed by national laws, and they are referred to differently in different European Member States [20], resulting in unfair competition between stakeholders. A common agricultural practice is fertilization. In this regard, microalgae are regarded as an organic fertiliser, as they can gradually release N, P, and K to prevent nutrient losses [21,22]. Furthermore, microalgae can recover nitrogen and phosphorus from wastewater by concentrating these nutrients in their biomass. There are various organic and inorganic compounds in Arthrospira spp., suggesting that this microalga could be used as a biofertilizer. The soil reservoir can be depleted of nutrients when crops are intensively produced. There are also some nutrients that can be affected by soil conditions, such as pH, salinity, calcium carbonate (CaCO3), and antagonism between macronutrients and micronutrients [23,24,25]. Besides macronutrients, micronutrients are also essential for crop growth and development [26]. As coenzymes for a variety of processes in plant metabolism, micronutrient deficiencies can cause severe problems in cell activities, resulting in non-optimal growth and decreased crop yields and quality [23,27]. Studies have shown that microalgae as biofertilisers increase nutrient uptake, biomass accumulation, and crop yields [28,29,30]. As a result of their potential applications in various agricultural cropping systems, as well as to improve agricultural sustainability, different studies were conducted to assess the effects of MBF on various crops, including rice, garlic (Allium sativum L.), fenugreek (Trigonella foenum-graecum L.), and tomato [31,32,33].

  1. La Torre, A.; Battaglia, V.; Caradonia, F. An overview of the current plant biostimulant legislations in different European Member States. J. Sci. Food Agric. 2016, 96, 727–734.
  2. Coppens, J.; Grunert, O.; Van Den Hende, S.; Vanhoutte, I.; Boon, N.; Haesaert, G.; De Gelder, L. The use of microalgae as a high-value organic slow-release fertiliser results in tomatoes with increased carotenoid and sugar levels. J. Appl. Phycol. 2016, 28, 2367–2377.
  3. Mulbry, W.; Kondrad, S.; Pisarro, C. Biofertilisers from algal treatment of dairy and swine manure effluents. J. Veg. Sci. 2007, 12, 107–125.
  4. Shaaban, M.M.; El-Fouly, M.M.; Abdel-Maguid, A.A. Zinc-Boron relationship in wheat plants grown under low or high levels of calcium carbonate in the soil. Pak. J. Biol. Sci. 2004, 7, 633–639.
  5. Shaaban, M.M.; Loehnertz, O.; El-Fouly, M.M. Grapevine genotypic tolerance to lime and possibility of chlorosis recovery through micronutrients foliar application. Int. J. Bot. 2007, 3, 179–187.
  6. Shaaban, M.M.; Hussein, M.M.; El Saady, A.M. Nutritional status in shoots of barley genotypes as affected by salinity of irrigation water. Am. J. Plant Physiol. 2008, 3, 89–95.
  7. Marschner, H. Mineral Nutrition of Higher Plants, 2nd ed.; Academic Pres: Boston, MA, USA, 1995.
  8. El-Fouly, M.M.; Shaaban, M.M.; El-Khdraa, T.F. Soil and plant nutritional status in fruit orchards in Syria. Acta Agron. Hung. 2008, 56, 363–370.
  9. Shaaban, M.M. Nutritional status and growth of maise plants as affected by green microalgae as soil additives. J. Biol. Sci. 2001, 6, 475–479.
  10. Shaaban, M.M. Green microalgae water extract as foliar feeding to wheat plants. Pak. J. Biol. Sci. 2001, 4, 628–632.
  11. Faheed, F.A.; Fattah, Z.A. Effect of Chlorella vulgaris as Biofertiliser on Growth Parameters and Metabolic Aspects of Lettuce Plant. J. Agric. Soc. Sci. 2008, 4, 165–169.
  12. Tarraf, S.A.; Talaat, I.M.; El-Sayed, A.E.K.B.; Balbaa, L.K. Influence of foliar application of algae extract and amino acids mixture on fenugreek plants in sandy and clay soils. Amino Acids 2015, 16, 19–58.
  13. Paudel, Y.P.; Pradhan, S.; Pant, B.; Prasad, B.N. Role of blue green algae in rice productivity. Agric. Biol. J. N. Am. 2012, 3, 332–335.
  14. Shalaby, T.A.; El-Ramady, H. Effect of foliar application of bio-stimulants on growth, yield, components, and storability of garlic (Allium sativum L.). Aust. J. Crop Sci. 2014, 8, 271.

  1. Accepted

  1. Anthrospira is not listed in the box of studied mixroalgae being commercially used and it rised question.

I have changed figure 1 to experimental microalgae and microalgae in trial that are already commercialised

  1. I don't mind providing general idea, but don't mix in one direct and indirect effecs of microalgae.

I have changed the order of the sentences within the paragraph to better divide the direct actions of microalgae from those that occur indirectly on plants

It is important to remember that the concentration of the biostimulant and the number of applications are a determining factor in the success of the treatment and that an increase in the dose does not always correspond to an increase in the positive effects on the plant [81]. In fact, it has been found in some studies that intermediate dilutions of biostimulant may be more effective in promoting growth and flowering, while the application of high doses usually reduces or even neutralises the effect. Effective doses may vary considerably depending on the plant species treated and the method of application. In general, foliar application is effective at lower concentrations than seed or soil application [59].In rice plants inoculated with microalgae, an accumulation of phenolic acids and flavonoids has been observed in leaf tissue [60]. In addition, according to recent studies, polysaccharide extracts obtained from different microalgal strains, including Chlamydomonas reinhardtii, Chlorella rokiniana, Porphirydium spp. and Dunaliella salina, can increase the activity of antioxidant enzymes such as catalase, peroxidase and superoxide dismutase in tomato plants subjected to salt stress [72]. Another important function of exopolysaccharides released into the soil is to sequester metal ions and sodium ions, reducing their uptake by plants and stimulating their growth in saline or polluted soils [78,79]. For example, coating maize seeds with Arthrospira platensis led to a reduction of more than 90% of the cadmium absorbed by the roots 12 days after sowing [80].

  1. It relates to higer water content in plants due to better root work (better water relations), but not water availability (in substrate).

I have added in the caption of figure 2 all the explanations related to the photo, with reference also to the root system where the algae stimulate the growth of the roots with a related increase in water and nutrient uptake.

Round 3

Reviewer 2 Report

The answeres are accepted.